# The impact of general practitioners working in or alongside emergency departments: a rapid realist review

Alison Cooper,[1] Freya Davies,[1] Michelle Edwards,[1] Pippa Anderson,[2] Andrew Carson-Stevens,[1] Matthew W Cooke,[3] Liam Donaldson,[4] Jeremy Dale,[3] Bridie Angela Evans,[5] Peter D Hibbert,[6,7] Thomas C Hughes,[8] Alison Porter,[5] Tim Rainer,[1] Aloysius Siriwardena,[9] Helen Snooks,[5] Adrian Edwards[1]

For numbered affiliations see end of article.

**Correspondence to**
Dr Alison Cooper;
coopera8@cardiff.ac.uk

## ABSTRACT

**Objectives** Worldwide, emergency healthcare systems are under intense pressure from ever-increasing demand and evidence is urgently needed to understand how this can be safely managed. An estimated 10%–43% of emergency department patients could be treated by primary care services. In England, this has led to a policy proposal and £100 million of funding (US$130 million), for emergency departments to stream appropriate patients to a co-located primary care facility so they are 'free to care for the sickest patients'. However, the research evidence to support this initiative is weak.

**Design** Rapid realist literature review.

**Setting** Emergency departments.

**Inclusion criteria** Articles describing general practitioners working in or alongside emergency departments.

**Aim** To develop context-specific theories that explain how and why general practitioners working in or alongside emergency departments affect: patient flow; patient experience; patient safety and the wider healthcare system.

**Results** Ninety-six articles contributed data to theory development sourced from earlier systematic reviews, updated database searches (Medline, Embase, CINAHL, Cochrane DSR & CRCT, DARE, HTA Database, BSC, PsycINFO and SCOPUS) and citation tracking. We developed theories to explain: how staff interpret the streaming system; different roles general practitioners adopt in the emergency department setting (traditional, extended, gatekeeper or emergency clinician) and how these factors influence patient (experience and safety) and organisational (demand and cost-effectiveness) outcomes.

**Conclusions** Multiple factors influence the effectiveness of emergency department streaming to general practitioners; caution is needed in embedding the policy until further research and evaluation are available. Service models that encourage the traditional general practitioner approach may have shorter process times for non-urgent patients; however, there is little evidence that this frees up emergency department staff to care for the sickest patients. Distinct primary care services offering increased patient choice may result in provider-induced demand. Economic evaluation and safety requires further research.

**PROSPERO registration number** CRD42017069741.

### Strengths and limitations of this study

► A realist approach to evidence synthesis leads to theory development that explains how and why context links to outcome; contextual factors can then be incorporated into the evidence base to inform healthcare management and policy-making.

► We used experts and stakeholders to facilitate the process, help confirm findings and produce a context-specific document in response to emerging issues.

► Some studies did not describe how general practitioners worked in adequate depth to identify key mechanisms that led to the outcomes.

► We have focused on general practitioners treating patients in emergency department settings relevant to the UK healthcare system; patient demographics and other healthcare professionals working in primary care services may vary and influence the effectiveness of these services.

## BACKGROUND

Worldwide, emergency healthcare systems are under intense pressure from ever-increasing demand.[1] Evidence is urgently needed to understand how best to manage this demand while safely achieving the highest standards of care.[2] An estimated 10%–43% of patients attending hospital emergency departments could be treated in primary care settings.[3–9] In England, this has led to a policy proposal, supported by £100 million of funding (US$130 million), that all emergency departments have a co-located primary care facility, so they are 'free to care for the sickest patients'.[10–12]

The UK has a universal healthcare system, the National Health Service (NHS), funded though taxation.[13] Primary care is led by general practitioners, community-based doctors with generalist training. General practitioners are described as working in or

alongside emergency departments in three main ways: treating patients identified as having primary care type problems in a unit alongside the emergency department including walk-in centres, urgent care centres or out-of-hours services; treating patients inside the emergency department, which may include patients presenting with a wider range of conditions; or working at the front door of the emergency department, redirecting patients with primary care type problems to an alternative primary care service off-site (including pharmacists, opticians or back to their own general practitioner).[14] There is little research evidence to guide decisions about how general practitioners most effectively work within these service models. The risk of provider-induced demand, potential patient safety issues and how to recruit a workforce for this initiative are also unclear.[15–19] Due to this uncertainty, the main standard-setting body of the NHS (National Institute for Health and Care Excellence) does not currently recommend general practitioners work in emergency department settings.[20]

Research studies addressing these questions are heterogeneous and few are conducted at scale.[15–17] This limits the results of traditional synthesis methods to shape practice or policy. Realist methods offer an alternative approach, generating theories to explain why a particular intervention is likely to work, how, for whom, in what circumstances and why.[21] These methods identify the important contextual factors that facilitate or inhibit desired intervention outcomes to inform healthcare management and policy-making.[22] Urgent and emergency care settings vary in geographical location, the type of patients, the presenting conditions and the experience and disciplines of the healthcare professionals that treat them. We decided that a realist approach, aiming to explain how general practitioners work in or alongside different emergency department settings and why the resultant successes or failures occur, would be more informative than a traditional review approach.

Our research question was, 'Why and how do general practitioners working in or alongside emergency departments affect: patient attendance and flow; patient experience; patient safety; and the wider healthcare system?'

## METHOD

We followed the realist review methodology to identify mechanisms (M) that explain how or why contexts (C) relate to outcomes (O), to generate theories described as context–mechanism–outcome configurations.[21] (Specific terminology is defined in table 1.) Our focus was specifically on general practitioners working in or alongside emergency departments. We used the rapid realist review approach described by Saul *et al.,* which uses experts and stakeholders, to streamline the process and to produce a context-specific product that is useful to policy-makers and responsive to emerging issues; providing evidence and making explicit what is known on the given topic, also articulating the current research gaps.[23] We registered our protocol on the PROSPERO database (http://www.crd.york.ac.uk/PROSPERO/display_record.php?ID=CRD42017069741) and followed RAMESES publication standards for realist reviews.[24] The period of study was April to November 2017.

Three reviewers (AC, FD and ME) conducted a scoping exercise with the four UK papers identified in the review by Ramlakhan *et al*[4 17 25–27] and two policy documents,[14 28] to generate initial theories. We then developed and piloted data extraction forms. Our theories were developed at the micro-level (the reasoning processes of general practitioners, emergency department staff and patients), meso-level (staff interactions resulting in department level outcomes) and macro-level (the impact on the wider system).[29]

We discussed these initial theories with the wider study team of 18 collaborators, including emergency department clinicians, policy-makers, general practitioners, members of the public and methodologists at a study meeting in May 2017. We used them as an expert reference group, to contribute ideas for other possible

| Table 1 | Glossary of terms |
|---|---|
| Primary care type problem | A condition that a typical general practitioner in a typical general practice would be expected to manage. |
| Streaming | A system, following brief clinical assessment, to allocate patients to the most appropriate healthcare provider within the emergency department setting.[122] |
| Triage | Identifying acuity and prioritising patients on that basis.[122] |
| Redirection | 'Sending people away' to an appropriate off-site or separately managed service.[122] |
| Context (C) | Pre-existing conditions which influence the success or failure of different interventions or programmes.[21 123] |
| Mechanism (M) | The intervention and people's reaction to it; how does it influence their reasoning?[21 123] |
| Outcome (O) | Intended and unintended results as a result of a mechanism operating within a context.[21 123] |
| Initial theory | An early theory informed by available evidence describing why, how and for whom the intervention is thought to work using a context–mechanism–outcome configuration.[21 123] |
| Refined theory | An initial theory that has been refined using primary or secondary evidence.[21 123] |

initial theories and to identify further research papers in peer-reviewed journals and relevant reports in the grey literature. Six members of this group (AP, PA, BAE, BH, JD and ACS) met via teleconference every 6 weeks to discuss findings and guide priority search areas.

We used papers referenced in three previous systematic reviews as a starting point,[15–17] and to identify papers published since, we combined search terms used previously.[16 17] A combination of free text and Medical Subject Headings terms was used (see online supplementary file 1 for Medline strategy which was adapted for other databases). AC ran the searches on the following databases from 15 June to 4 July 2017: Medline via OVID, Embase, CINAHL, Cochrane DSR & CRCT, DARE, HTA Database, Business Source Complete, PsycINFO and SCOPUS and used EndNote X8 (Clarivate analytics) to export citations from the database searches and identify duplicates. AC screened the titles and abstracts of all identified papers using a checklist, developed and tested in collaboration with FD, which ranked abstracts according to relevance.

We selected studies if they could contribute to the process of theory development at the level of individual data extracts rather than assessing the full text against a set checklist.[24 30] We excluded papers that lacked relevance or explanatory power, or were unavailable in English. AC and FD imported data extracts into NVivo V.11 (QRS international) that evidenced how mechanisms (M), influenced by local contexts (C), related to outcomes (O). Quantitative, qualitative or contextual data were extracted from any part of a paper. We continually considered the relevance and rigour of each included piece of evidence during the data extraction and synthesis phases.[30] We discussed weekly within the team (AC, FD, ME and AE) how individual data extracts should be used to ensure appropriate inferences were made.[30] A quarter of all included articles was read by both reviewers, and the coding process was discussed in detail, to ensure consistency of approach.

We used snowballing techniques (such as searching companion papers and citation tracking) for all included articles. We also searched to identify additional relevant grey literature (including policy documents and opinion pieces) from a variety of sources. The search process was iterative, overlapping with data extraction and analysis, and was directed towards the evidence gaps and finding explanatory information.

We applied Pawson's reasoning processes,[21] to synthesise the evidence and develop our theories. We presented these context-specific developing theories to our expert reference group in November 2017. At this stage, the group recognised that although the review had been useful in theory development, there were limited opportunities for theory testing and refinement due to evidence gaps. Rather than continuing to search the literature, we decided that gathering primary data from our evaluation case study sites in the next phase of our wider ongoing study,[31] would give an opportunity for more meaningful testing to derive refined theories.[21]

**Patient and public involvement**

Three public contributors (BAE, BH and JH) were co-applicants for the funded research and contributed to the conceptualisation of our wider study, including theory generation through the review.[31] They contributed in both meetings described above to ensure that the patient's perspective was acknowledged and at a stakeholder dissemination event in February 2018.

## RESULTS

Figure 1 shows the search strategy and results. A total of 96 articles contributed to the developing theories. The articles were largely primary research studies, most from the UK (n=44 articles), with a large contribution from the Netherlands (n=17). Others were from Ireland, Belgium, Switzerland, Sweden, Italy, Finland, Australia, USA, Canada, Singapore and New Zealand. Most described patients identified by the emergency department as having primary care type problems, appropriate for treatment by a general practitioner.

We synthesised data to develop theories, described using Context (C)–Mechanism (M)–Outcome (O) configurations, to explain: how or why emergency department staff and general practitioners react to guidance to determine which patients are streamed to general practitioners; the role general practitioners may adopt in the emergency department setting (traditional general practitioner, extended general practitioner, gatekeeper or emergency clinician); and how these factors influence patient (experience and safety) and organisational (risk of provider-induced demand and cost-effectiveness) outcomes. These theories are summarised in table 2 with an indication of supporting data. Full details of included articles are listed in online supplementary file 2.

### Theory 1: Effectiveness of the streaming system

*General practitioners and emergency department staff use their own personal experience and expectation (C) when interpreting streaming guidance (M) to influence which patients are streamed to general practitioners (O).*[4 14 25 32–40]

Twelve articles supported this theory and indicated how the streaming process itself directly influenced the effectiveness of the general practitioner service in the department. Variable streaming rates were described due to differences in guidelines and also how the guidance was interpreted by emergency department clinical and non-clinical staff of varying experience.[32 37 38 40 41] The (streaming) nurse was sometimes described as being unclear which patients general practitioners could deal with,[4 25 34–38] or being more familiar with emergency department work so favouring emergency department referral,[14 33 35 37–39] even overruling the guidelines if he/she felt that the patient would require specific investigations,[35] or admission.[33] General practitioners were also noted to over-ride nurse decisions to select patients that suited their own interests or perceived skills.[42] Increased general practitioner streaming rates were reported when

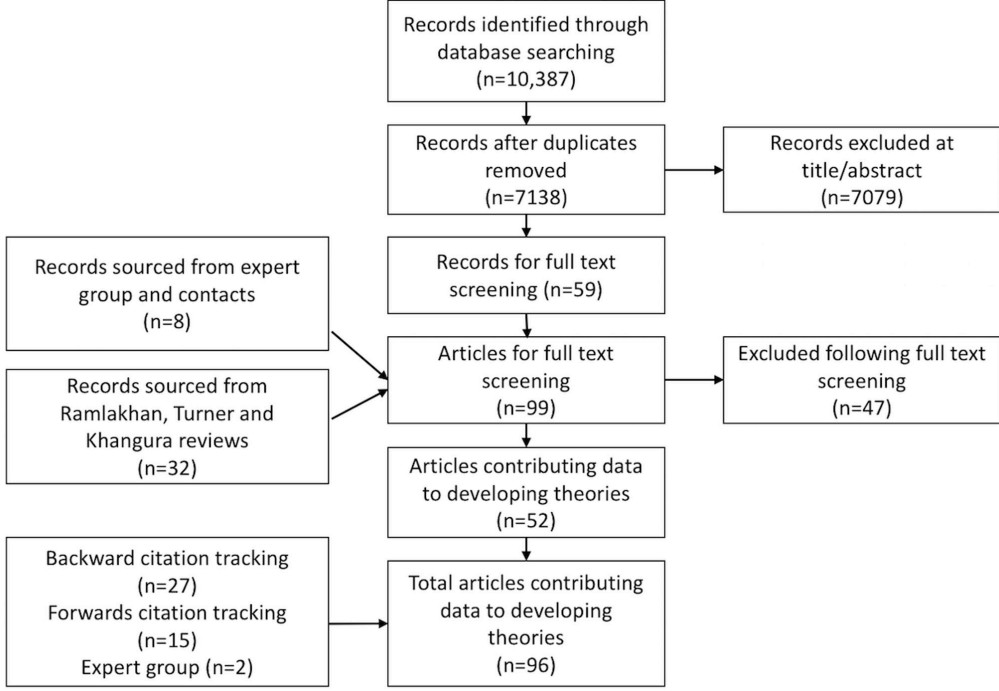

**Figure 1** Search strategy and results.

there was a good relationship between the general practitioners and emergency department nurses,[40] and when the general practitioners were directly involved in the streaming process.[43 44] The influence of commissioning or leadership was not described.

### Theory 2a: Traditional general practitioner role versu*s* emergency clinician role

*When general practitioners working in the emergency department maintain a 'traditional role' using the same approach taken in the primary care setting (M) to treat patients with primary care problems (C),[38 39 45–48] investigations, admissions and process times will reduce (O).[4 5 25 26 45 47–52] However, if general practitioners adopt an 'emergency clinician role' working as another pair of hands ('going native') because of their personal interest or experience or because they feel this is the correct way to work in this setting (M), there will be no difference in the rate of investigations and admissions (O).[38 41]*

The traditional general practitioner approach was described by many authors as a different approach to risk management and diagnostic uncertainty, with less reliance on acute investigations.[38 39 46–48] This approach was maintained in a variety of different settings despite full access to investigations—when general practitioners were allocated a separate consulting room mimicking usual general practice,[4 25] and also when general practitioners worked in a more fully integrated model, alongside emergency department clinicians.[45 47 48] Other articles reported general practitioners managing non-urgent patients in this way to divert attendances from emergency department staff.[32 36 37 43 44 53–65]

There were limited qualitative data to support the 'emergency clinician role' theory.[38] An Irish study described

an 'unstructured receptionist-based triage system' for all patients attending the department (including referrals from primary care) which may have influenced relatively inexperienced general practitioners to adopt a 'diagnosis driven' emergency clinician approach.[41] The influence of general practitioners' special interests, experience in emergency medicine or the effect of staff shortages were not described in the literature to affect this potential role shift.

### Theory 2b: Extended general practitioner role

*General practitioners in emergency departments can work in an 'extended role' where their skills are directed at specific patient groups including non-urgent paediatric or elderly patients (C) to treat using the usual primary care approach (M) to reduce the use of hospital resources and admissions in these patient groups (O).[5 28 43 53 66]*

Several paediatric primary studies supported general practitioners treating children triaged as 'non-urgent' to divert attendances from the emergency department,[43 53] and reduce hospital admissions.[5 66] None of the included primary studies described general practitioners specifically treating care home residents or the elderly, as suggested in a policy document.[28]

Smith *et al* reported an increase in antibiotic prescribing for children by general practitioners,[5] which could potentially be an unintended consequence of the 'traditional role' approach; relying on clinical acumen and treating a suspected source of infection rather than admitting, investigating and observing the patient to confirm the diagnosis. An increase in prescribing by general practitioners was not described in other UK studies,[4 25] but was

**Table 2** Summary of developing theories and supporting evidence

| Theory | Context (C)–Mechanism (M)–Outcome (O) configuration | Example of supporting extract | Evidence base |
|---|---|---|---|
| 1. Effectiveness of the streaming system | General practitioners (GPs) and emergency department (ED) staff use their own personal experience and expectation (C) when interpreting streaming guidance (M) to influence which patients are streamed to GPs (O). | 'It seems that patients are difficult to classify (for A&E (ED) or walk-in centre GPs or nurse practitioners) on limited information for several reasons: serious conditions can sound minor, and vice versa; conditions can present in various ways; and complaints can have several underlying causes.'[35] | Data to support theory. [4 14 25 32–40] |
| 2a. Traditional general practitioner (GP) role versus emergency clinician role | When GPs working in the ED maintain a 'traditional role' using the same approach taken in the primary care setting (M) to treat patients with primary care problems (C), investigations, admissions and process times will reduce (O). However, if GPs adopt an 'emergency clinician role' working as another pair of hands ('going native') because of their personal interest or experience or because they feel this is the correct way to work in this setting (M), there will be no difference in the rate of investigations and admissions (O). | 'I guess our emergency medicine approach is we're looking for something dreadful and a GP approach is very different in that most of the time they know it's minor stuff or … moderate stuff that is self-limiting and so … they're looking to find symptomatic relief and how can we get this patient home and away from hospital.' (Consultant)[38]<br><br>'Once they start becoming like everyone else then they stop being like a GP and they don't necessarily work quickly and effectively which is supposed to be the whole benefit of having them there.' (Consultant)[38] | Data to support traditional GP role theory [4 5 25 26 38 39 45–52]<br><br>Limited data to support ED clinician role theory.[38 41] |
| 2b. Extended GP role | GPs in EDs can work in an 'extended role' where their skills are directed at specific patient groups including non-urgent paediatric or elderly patients (C), to treat using the usual primary care approach (M), to reduce the use of hospital resources and admissions in these patient groups (O). | 'During a 6-month pilot scheme which co-located a primary care GP service in a busy paediatric ED, patients seen during the hours when the GP was available were significantly less likely to be admitted, exceed the 4 hours waiting target or leave before being seen, but more likely to receive antibiotics.'[5] | Data to support theory for paediatric patients only. [5 28 43 53 66] |
| 2c. Gatekeeper role | GPs can use their generalist skills and knowledge of community resources (M) to redirect patients presenting with primary care problems (C) out of the ED to alternative primary care services off-site for treatment thereby reducing ED attendances (O). | 'GPs and nurses based in triage identify patients who could be managed more appropriately in primary care as soon as they enter the ED, and redirect them back to primary care services.'[70] | Limited data to support theory. [70 71] |
| 3. Patient satisfaction | Patients with primary care problems that present to EDs (C) and are seen by GPs, are more satisfied with the care they receive (O) if the experience exceeds expectation (M), but if they do not perceive any difference in the care they received compared with what they expected (M), there is no difference in satisfaction (O). | 'There were no significant differences in (patient) satisfaction ratings between the three groups of doctors (GPs, Senior House Officers or Registrars).'[39] | Limited data to support theory. [26 39 45 47 86–90] |

Continued

**Table 2** Continued

| Theory | Context (C)–Mechanism (M)–Outcome (O) configuration | Example of supporting extract | Evidence base |
|---|---|---|---|
| 4. Safety implications | In EDs where there are delayed patient transfers to wards or inadequate staffing (C) GPs seeing patients with primary care type problems (M), may not free up ED staff to care for the sickest patients (O). | 'The attribution of overcrowding in ED to attendance by GP-type patients is simplistic; it does not address how patients are processed within ED or how they are transferred to wards later if required.'[46] | Limited data to support theory. [46 93–99] |
| 5. Risk of provider-induced demand | If patients with primary care type problems present to EDs (C) and are streamed to indistinct primary care services, without patient awareness or choice (M), there is no provider-induced demand (O). However, distinct urgent primary care services may offer convenient access to primary care (M), resulting in provider-induced demand (O). | 'A&E (ED) has not seen any reduction in their patients. If there is a service, patients will use it. You could have three walk-in centres in the city and all three would be used and you may still not see any dropping in A&E (ED) counts.' (Manager)[102] | Data to support theory. [4 27 54–56 80 102–109] |
| 6. Cost-effectiveness | If there is demand for patients with primary care problems presenting to EDs (C), and they are streamed to on-site GPs and managed using a traditional GP approach (M), the service is cost-effective due to fewer referrals, admissions, investigations and better outcomes compared with usual services (O). | 'Management of patients with primary care needs in the A&E department by GPs reduced costs with no apparent detrimental effect on outcome.'[26] | Limited data to support theory. [26 45 51] |

reported (but not the drugs involved) in both Irish studies that involved more junior general practitioners.[41 45]

There was evidence that general practitioners working in or alongside emergency departments see a different cohort of patients to that in usual general practice, with more acutely unwell patients,[38 67] and minor injuries,[4 6 35–38 68 69] which could also be described as an 'extended role.' There was no evidence in the included studies for the implications of this on their skillset, learning needs, cognition processes or risk management behaviour.

### Theory 2c: Gatekeeper role

*General practitioners can use their generalist skills and knowledge of community resources (M) to redirect patients presenting with primary care problems (C) out of the emergency department to alternative primary care services off-site for treatment thereby reducing emergency department attendances (O).*[70 71]

There were limited data to support this theory with two London case study reports identified in an 'accident and emergency avoidance scheme' document, describing 228 patients in total.[70 72] There was evidence that general practitioners were more likely to redirect patients after an initial assessment than senior emergency department nurses, but only from a sample of 384 patients that self-presented to a London emergency department.[71]

Due to a lack of evidence for general practitioners performing a redirection role, following realist methodology, we also included studies involving redirection of patients from the emergency department by a senior emergency department clinician or nurse to gain understanding about how and why the system worked. Many of these articles described reduced emergency department attendances.[73–79] Previous UK guidance has cautioned against redirecting patients from emergency departments due to the risk of delayed assessment and treatment, especially in vulnerable patient groups including the homeless or those with mental health problems who may not go on to receive the care they need.[14 28] Studies from Scotland, Sweden and the USA that described a comprehensive assessment process, including measurement of vital signs and a focused history, reported that their redirection policies were safe and worked well to reduce attendances.[74 76 78–80] Other US studies, that did not describe the assessment process, reported adverse events when children were redirected without treatment.[81 82] The low sensitivity of triage criteria to identify those that needed urgent care,[83] especially infants[84] and failure to validate a predictive model for refusal of care,[85] were highlighted in other studies. The influence of governance processes restricting redirection of patients by some staff to services off-site was not described in these articles.

### Theory 3: Patient satisfaction

*Patients with primary care problems that present to emergency departments (C) and are seen by general practitioners, are more satisfied with the care they receive (O) if the experience exceeds expectation (M), but if they do not perceive any difference in the*

*care they received compared with what they expected (M), there is no difference in satisfaction (O).*[26 39 45 47 86–90]

Data to support this theory were limited, with an increase in satisfaction by patients seen by general practitioners generally associated with shorter waiting times,[47 86] rather than expectation of investigation and treatment.[39] The general practitioners were sometimes supernumerary which may have contributed towards this.[26 47] Other studies demonstrated that general practitioners focused more on patient education and counselling than emergency department clinicians with some improvement in satisfaction rates.[91 92] In more fully integrated models, the patient was often unaware that they had seen a general practitioner rather than an emergency department clinician and there was no difference in patient satisfaction.[26 39 45 87]

### Theory 4: Safety implications

*In emergency departments where there are delayed patient transfers to wards or inadequate staffing (C) general practitioners seeing patients with primary care type problems (M), may not free up emergency department staff to care for the sickest patients (O).*[46 93–99]

There was a lack of evidence that general practitioners working in or alongside emergency departments directly or indirectly improved care and safety for the sickest patients. A reduction in time spent in the department for patients requiring emergency department level care was suggested in a UK simulation and modelling study,[93] and an Australian study also reported a reduced mean time taken to see more seriously ill patients but this was not seen on sites that described provider-induced demand.[94] A Canadian study of over 4 million patient visits reported that low complexity emergency department patients did not increase time to first physician contact for high-complexity patients.[95] Other studies also described how diverting non-urgent patients did not improve the high-level care required by others, and that influences such as delayed transfer of patients to the ward were more likely to contribute to overcrowding.[46 96–98] Staffing levels, staff attitude and the time of day were independent factors described to affect emergency department flow.[99]

There were minimal data on the safety implications of general practitioners working in emergency department settings. Several studies used emergency department re-attendance as a marker of safety, with no increase among patients seen by general practitioners compared with usual emergency department staff.[26 27 45 100 101] Annual death rates were used as another crude marker in a Dutch study, with no significant increase following the introduction of an out-of-hours primary care physician cooperative.[55] Shared or separate governance systems between general practitioners and the emergency department were rarely described in the primary studies, providing no evidence for best practice. For general practitioners working inside the emergency department good communication and integration were described in some

studies,[4 26 38 67] with anecdotal reports of poor communication negatively affecting care quality in others.[32]

### Theory 5: Risk of provider-induced demand

*If patients with primary care type problems present to emergency departments (C) and are streamed to indistinct primary care services, without patient awareness or choice (M), there is no provider-induced demand (O).*[4 27 54 55] *However, distinct urgent primary care services may offer convenient access to primary care (M), resulting in provider-induced demand (O).*[56 80 102–109]

Four articles described fully integrated models, where non-urgent patients were streamed directly to general practitioners inside the emergency department without provider-induced demand.[4 27 54 55] Here, there was no patient choice offered and often a lack of patient awareness. Another 10 articles described distinct urgent primary care services, often in separate buildings outside the emergency departments, as duplicating services and creating their own demand, increasing patient presentation rates directly or at nearby services, rather than relieving pressure on the emergency department.[80 102–112]

### Theory 6: Cost-effectiveness

*If there is demand for patients with primary care problems presenting to emergency departments (C), and they are streamed to on-site general practitioners and managed using a traditional general practitioner approach (M), the service is cost-effective due to fewer referrals, admissions, investigations and better outcomes compared with usual services (O).*[26 45 51]

Data to support this theory were limited, but supported by three economic evaluations (UK, Ireland and the Netherlands) where non-urgent patients were streamed to general practitioners during normal daytime hours.[26 45 51] The comparator was 'business as usual' with no general practitioner service. The UK and Irish studies were published in 1996 and may not represent current emergency department staffing models. No articles were identified that studied the relative cost-effectiveness of general practitioners redirecting patients from the emergency department for care elsewhere. A 5-year US redirection study calculated cost-effectiveness from the perspective of the institution but did not include costs for treatment incurred elsewhere,[76] while another US study calculated that marginal costs for non-urgent visits to the emergency department were low and that cost savings from diverting visits may be less than widely believed.[113] However the USA has a complex health system, with a significant majority of the population covered by private health insurance alongside state-funded Medicare, Medicaid, the federally funded Veterans Health Administration, and a substantial uninsured population—all factors which could influence access to emergency departments and the type of care needed and delivered.

Three other studies of 'out-of-hours' patients did not find the addition of a primary care service to be cost saving. One Dutch study, with an off-site general practitioner cooperative, reported parents refusing to take their children to a different location, or the (streaming)

nurse overruling the policy.[33] Another 12-year-old Dutch study showed no change in costs, despite a substantial reduction in emergency department attendances, due to regulations dictating minimum staffing levels to cope with major trauma.[65] The Dutch healthcare system has a complex funding structure with a mix of social and private insurance and this may influence incentives and disincentives to access emergency departments. An Australian primary care out-of-hours service closed because patients chose to attend an equally accessible general practice service that existed nearby.[40]

### Wider system implications

Limited evidence from the included studies prevented us from developing theories on wider system implications. There were no reports of emergency department clinicians being encouraged to adopt a more conservative approach, as a result of working alongside general practitioners, but some reports of general practitioners in management positions influencing system changes.[114 115] The potential reduction in learning opportunities for junior doctors was highlighted in two articles.[52 67] There was limited evidence that working in an emergency department setting led to increased job satisfaction for some UK general practitioners with a special interest in emergency care.[38 115] However, reduced satisfaction was also described because the job was outside the scope of usual general practice,[38 50] possibly contributing towards recruitment problems.[38 116]

## DISCUSSION
### Principal findings

We developed theories using data from 96 articles to describe the mechanisms by which general practitioner services are linked to outcomes: about the streaming process itself; the role general practitioners may adopt in the emergency department setting; and the effects of these on the patient (experience and safety) and the organisation (risk of provider-induced demand and costs). There was little evidence that general practitioners in emergency departments directly or indirectly affected the care and throughput of the sickest patients. Distinct units, advertising these services, may offer an attractive alternative to primary care and result in provider-induced demand. The literature describing economic impacts of general practitioners in emergency departments comes from different countries, with different funding systems and spans over 20 years, limiting conclusions.

### Strengths and limitations

Heterogeneous studies involving general practitioners working in or alongside emergency departments do not suit traditional systematic review methods. We have conducted the first realist review in this area, using methods that are gaining prominence in healthcare research.[117 118] The rapid realist review approach is appropriate in relation to the rapidly evolving NHS policy on emergency department use of general practitioners,[10–12] showing where such policies may be reinforced or refuted by the evidence available.[23] A weakness of our study was the time constraint on our project but the expert group mitigated this, and enabled us to focus and direct our research.[23] Some studies did not describe the intervention in adequate depth to help facilitate the identification of key mechanisms. Single-site heterogeneous studies and the nature of different healthcare and funding systems limited international comparability.[21]

The wide estimates of patients presenting with primary care type problems to emergency departments highlight the difficulty in defining and identifying this target patient group and therefore the effectiveness of these services in different local contexts. We have focused on general practitioners working in or alongside emergency departments but in the UK this role has evolved to include nurses and advanced care practitioners from other disciplines, often due to staffing and recruitment challenges. These challenges may be mirrored in emergency department-based services, affecting variation between services and need to be considered in further research.

### Comparison with other reviews

Before our review, the largest review to date by Ramlakhan et al.[17] included 20 papers and described provider-induced demand, poor evidence for improved emergency department throughput and minimal economic impact.[17] The Goncalves-Bradley et al. Cochrane review of four studies, published in 2018, highlighted inconsistent results and a lack of evidence on safety.[18] We also found evidence of provider-induced demand in distinct primary care units but less so in more fully integrated service models where patients lacked awareness that they had been directed to primary care services.[4 54 55] We found that patients with primary care problems may have reduced process times if treated by general practitioners adopting a traditional role but there was a lack of evidence for an improvement in overall throughput for patients in the department. There was also a lack of evidence on the impact on general practitioners' cognition processes and risk-taking behaviour when treating a different group of patients to that seen in usual general practice and the safety implications of this.

### Policy implications

The global health priority recently given to Universal Health Coverage,[119] and the attention being given to the 40th anniversary of the Alma-Ata declaration,[120] moves to centre stage the design of primary healthcare systems, particularly their capacity and capability to respond to urgent care needs. Internationally, emergency departments are exploring options on how to run more efficiently and safely. Our theories, informed by literature from 13 countries, allow policy-makers to make more considered judgements about their relevance to their own contexts for service provision. The UK has already commissioned further research in this area, funded by the

National Institute for Health Research (HS&DR Projects: 15/145/04[31] and 15/145/06[121]), the former collecting primary data to further test and refine these theories.

## CONCLUSION

The effectiveness of emergency department streaming to primary care services may be influenced by how staff interpret the streaming system and the roles general practitioners adopt. Caution is needed in embedding the policy until further research and evaluation are available. Service models that encourage the traditional general practitioner approach may have shorter process times for non-urgent patients; however, there is little evidence that this frees up emergency department staff to care for the sickest patients. Distinct primary care services offering increased patient choice may result in provider-induced demand. Economic evaluation and safety requires further research.

**Author affiliations**
[1]Division of Population Medicine, Cardiff University, Cardiff, UK
[2]Centre for Health Economics, Swansea University, Swansea, UK
[3]Warwick Medical School, University of Warwick, Coventry, UK
[4]London School of Hygiene and Tropical Medicine, London, UK
[5]College of Medicine, Swansea University, Swansea, UK
[6]Faculty of Medicine and Health Sciences, Macquarie University, Sydney, New South Wales, Australia
[7]University of South Australia Division of Health Sciences, Adelaide, South Australia, Australia
[8]Emergency Department, John Radcliffe Hospital, Oxford, Oxfordshire, UK
[9]School of Health and Social Care, University of Lincoln, Lincoln, UK

**Acknowledgements** Many thanks to our public contributors, Barbara Harrington and Julie Hepburn and to Damon Berridge for contributing to discussions. Also to Faris Hussain for assisting with citation tracking.

**Contributors** AC, FD, PA, ACS, MWC, LD, JD, BAE, PDH, TCH, AP, TR, AS, HS and AE are co-applicants on the wider project and were involved in the conceptualisation of the study. AC, FD, ME, PA, ACS, MWC, JD, BAE, PDH, TCH, AP, TR, AS, HS and AE contributed as part of the expert group in team meetings in May and/or November 2017. AC and FD planned the synthesis approach with input from AE. ME contributed to data analysis and interpretation in the pilot work and weekly team meetings. The core review team (AP, PA, BAE, JD and ACS) met via teleconference every 6 weeks to discuss findings and guide further searches. AC conducted the database searches. AC and FD extracted data extracts and were involved in the synthesis process, meeting weekly with AE and ME to discuss findings. AC prepared the first draft of the manuscript which was reviewed and critically appraised by all authors, who approved the final version and agree to be accountable for this work.

**Funding** This study is funded by the National Institute for Health Research (NIHR) HS&DR Project 15/145/04.

**Disclaimer** The views expressed are those of the authors and not necessarily those of the NIHR or the Department of Health and Social Care.

**Competing interests** None declared.

**Patient consent for publication** Not required.

**Provenance and peer review** Not commissioned; externally peer reviewed.

**Data sharing statement** No additional unpublished data are available.

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
