## [Reviewer comments · BMJ Open]

This paper was submitted to a another journal from BMJ but declined for publication following peer review. The authors addressed the reviewers' comments and submitted the revised paper to BMJ Open. The paper was subsequently accepted for publication at BMJ Open.

(This paper received three reviews from its previous journal but only two reviewers agreed to published their review.)

ARTICLE DETAILS

TITLE (PROVISIONAL)	The impact of general practitioners working in or alongside emergency departments: a rapid realist review
AUTHORS	Cooper, Alison; Davies, Freya; Edwards, Michelle; Anderson, Pippa; Carson-Stevens, Andrew; Cooke, Matthew W; Donaldson, Liam; Dale, Jeremy; Evans, Bridie Angela; Hibbert, Peter D; Hughes, Thomas C; Porter, Alison; Rainer, Tim; Siriwardena, Aloysius; Snooks, Helen; Edwards, Adrian

VERSION 1 – REVIEW

REVIEWER	Shammi Ramlakhan Sheffield Children's Hospital UK
REVIEW RETURNED	07-Jul-2018

GENERAL COMMENTS	Thank you for the opportunity to review your manuscript on this relevant and important topic. It addresses some important implications of service design of key aspects of the unscheduled care landscape, as well as on primary care provision and staffing in the future. The methodology is appropriate and well described, and is perhaps only limited by the quality of the current available evidence. Background/Discussion: The authors mention demand driving ED pressures in high income countries, however data suggests that this is also true in middle and lower income countries (LMICs), so perhaps the specification of high income is unnecessarily limiting at the outset. Similarly, recent UK reports suggest that traditional increased winter demand has now extended into other times of the year, with the concept of seasonal demand evolving to near-perennial higher levels of demand across all aspects of urgent care. Specifying winter demand perhaps does not add anything. The wide estimate of primary-care suitable patients suggests that this figure is not as straightforward to determine. Multiple confounding factors and variable (retrospective) analysis confirm that the issue is more complicated than it is viewed by policymakers and professional colleges. Although to be fair, this is
--

	not the focus of the manuscript, perhaps some discussion of the challenges in defining the target population is warranted, not least because this is critical to determining the cost-effectiveness and sustainability of any service which is premised on proportional or absolute reductions in demand/utilisation. Although specifically limited to GP delivered interventions, many primary care services are now staffed with a heterogeneous mix of GPs/nurses/AC practitioners, and this may further complicate the evaluation of services. Some of this evolution in skill-mix reflects staffing/recruitment challenges in traditional general practice, which is unsurprisingly mirrored in co-located GP type services. The authors appropriately highlight the dynamic nature of the services' development, but there are likely to be layers of intra- and inter- service variation as well. This makes like for like comparison particularly problematic. Results/Discussion: A key determinant of input and flow in GP services lies in the individual performing the "streaming" or selection - some studies/services use receptionists, nurses, ED doctors or GPs; the seniority/experience of these individuals, combined with flexibility of selection criteria (if any) impacts on the performance of the service. Some discussion of context as related to the type and seniority of staff involved in all steps of the streaming process is warranted. As the authors mention, this distinction is not always apparent in the primary research available, but it is nonetheless important. Similarly, the gatekeeper role should be interpreted with caution, as professional/institutional governance processes may restrict redirection to external services by nurses, therefore, as with much of the evidence, it may not be comparing like for like. With regards satisfaction, GPs are arguably more experienced in managing diagnostic expectation/uncertainty as related to patient satisfaction than (diagnosis driven) ED junior clinicians. This could also contribute to GPs' lower resource utilisation. The discussion and conclusions are appropriate, but may now need to take into account some of the points raised above. Minor: In Strengths and Limitations, 2nd sentence - "...this area, USING methods..."
--	--

REVIEWER	Suzanne Ablard (Research Associate) The University of Sheffield, United Kingdom
REVIEW RETURNED	09-Jul-2018

GENERAL COMMENTS	This is an interesting and topical paper, particularly given the government policy drive to have a co-located primary care service at all Emergency Departments in England. The authors have followed realist review methodology, using the international literature, to generate theories about how and why general practitioners working in or alongside emergency departments affect: patient attendance and flow; patient experience; patient safety; and the wider healthcare system. Whilst this study design is appropriate to answer the research
---

	question, I do have reservations over the broad use of the international literature to answer the research question. Health systems vary considerably across the world, particularly taking into consideration publicly funded versus privately funded health systems, and this will have an impact over the way in which co-located primary care services are implemented / operate. Therefore, there needs to be a greater appreciation of this in the write up of the study. For example, the background to the paper is strongly focused on the UK policy context with no mention of the international perspective. Page 7, line 31: “Six members of the group, including two” could be re-worded. Who are the two? I am uncertain about the relevance of figure 2, perhaps expand on what this means within the main text of the paper. Furthermore, the difference between “primary care type problems” and “low acuity” requires further explanation.
--	---

REVIEWER	A/P Robyn Cant Federation University Australia, Australia
REVIEW RETURNED	14-Nov-2018

GENERAL COMMENTS	Thank you for presenting this scholarly work: 'The effectiveness of primary care service models in or alongside emergency departments: a rapid realist review.' It provides a well written and well developed critique of the roles of general medical practitioners in regard to their working with or within emergency care departments. But the review is restricted to a subgroup of options for models of primary care that have been developed? I would like to make several comments and some suggestions for improvement. Please consider that descriptions need to be explanatory for the international readership – we are not in the UK system and so I suggest your further interpretation needs to be added in some parts. The title should be revised. The paper does not actually describe primary care service ‘models’ (GP Co-operatives, walk-in centres, after-hours centres, co-location or otherwise) you restrict your objective to describing GPs working “in or alongside” ED. Unclear- does this include a primary care clinic co-located in the same building as ED? (I think you said 'yes'. If you take the view that you describe general practitioners working ‘in or alongside”, this, then, rather ignores the body of literature about ‘models’ of care (although I note you list a number of these studies in your tables). Be wary of describing this as ‘GP services’ or ‘primary care services’ for this term remains unclear. Page 2 Abstract: please be more specific regarding design/content. I have posted some comments on the pdf of the submission. Suggest there is no intervention, please remove. Page 3 Abstract line 2-3 Conclusion: “Multiple factors influence the effectiveness of emergency department streaming to primary care services” should be amended to enable understanding that you refer to a general practitioner working within emergency department (not to the broader model of GP services allied to ED). Page 3 Abstract Line 9-10: you overstate claims?- “ there is little evidence on the safety implications or whether this improves care for the sickest patients.” Please amend as ‘the sickest patients’ are
---

beyond the scope of this review, your GP patient group will be lower acuity, less acutely ill patients?

Page 4 line 34: "Three primary care ..." - there is a gap here as you should state that of the patients presenting to ED for emergency care, a number are assessed as less urgent cases and may be triaged to GP for care? Please clarify.

Page 4 Line 37-46 please revise to clarify, see notes on PDF

P6 line2: your citations for Box 1: Glossary of terms(21,23) relate to design, not to definitions. Suggest there needs to be a more detailed definition of Triage and cite the UK policy. 'Streaming' seems to be confounded as I note citations describe a broader view of referral on than mere streaming (you say: "A system to allocate patients to the most appropriate healthcare provider within the emergency department setting"). But the text seems to refer to a broader view of the settings. Lower acuity patients are often referred on to a GP service from ED. This seems missing from your general text and it would be good to state several times that the types of pts that are managed by GPs would be in the lower range of triage 4.5 (pt type not mentioned but this is an important explanatory element)?

Page 7: your description of the process of synthesis is very helpful.

Page 12 line 12: please insert short summary of GP options in this case for streaming – ie, co-located in a clinic, embedded, or sit at front of dept to screen patients? As we are unaware. Please explain " due to the guidance itself".

Page 15 line 5 onwards: GP as gatekeeper role. Please add further description here. I am unaware that GPs may be at the forefront of an ED dept- however citation 14 gives some explanation of the context of this 'refer out' situation - that should be added. Is it just 'refer on'? ('UK' is a universal accepted abbreviation).

Page 21 line 18-30 limitations please summarize to reduce repeated text this was all stated in methods. Is not a limitation that you considered only a limited range of GP models- not walk-in centres for example.

While I have suggested additions be made to an already lengthy report, there may be some areas where you can reduce the wordiness?

Thank you for this comprehensive report. It is difficult to do justice to all your work however I feel that me being an outsider gives me a fresh view of your framework and reporting. Please see both my list of comments and comments on the pdf copy where the connection was more easily made. Please review punctuation there are too many unnecessary commas that break up the text and perhaps some commas where there should be semicolon? Please review reference list format to be more consistent with the referencing and use of capitalization, etc. I do consider that each reference needs to first give the source or author, however it is up to the publishers. Well done.

	The reviewer provided a marked copy with additional comments. Please contact the publisher for full details.
--	--

VERSION 1 – AUTHOR RESPONSE

Reviewer: 1

"Thank you for the opportunity to review your manuscript on this relevant and important topic. It addresses some important implications of service design of key aspects of the unscheduled care landscape, as well as on primary care provision and staffing in the future. The methodology is appropriate and well described, and is perhaps only limited by the quality of the current available evidence."

Thank you for this positive feedback.

Background/Discussion:

"The authors mention demand driving ED pressures in high income countries, however data suggests that this is also true in middle and lower income countries (LMICs), so perhaps the specification of high income is unnecessarily limiting at the outset. Similarly, recent UK reports suggest that traditional increased winter demand has now extended into other times of the year, with the concept of seasonal demand evolving to near-perennial higher levels of demand across all aspects of urgent care. Specifying winter demand perhaps does not add anything."

Thank you – amended to remove 'high income' and 'seasonal pressures' (page 5).

"The wide estimate of primary-care suitable patients suggests that this figure is not as straightforward to determine. Multiple confounding factors and variable (retrospective) analysis confirm that the issue is more complicated than it is viewed by policymakers and professional colleges. Although to be fair, this is not the focus of the manuscript, perhaps some discussion of the challenges in defining the target population is warranted, not least because this is critical to determining the cost-effectiveness and sustainability of any service which is premised on proportional or absolute reductions in demand/utilisation."

Expanded in the discussion section under limitations (page 25).

"Although specifically limited to GP delivered interventions, many primary care services are now staffed with a heterogeneous mix of GPs/nurses/AC practitioners, and this may further complicate the evaluation of services. Some of this evolution in skill-mix reflects staffing/recruitment challenges in traditional general practice, which is unsurprisingly mirrored in co-located GP type services. The authors appropriately highlight the dynamic nature of the services' development, but there are likely to be layers of intra- and inter- service variation as well. This makes like for like comparison particularly problematic."

Expanded in the discussion section under limitations (page 25).

Results/Discussion:

"A key determinant of input and flow in GP services lies in the individual performing the "streaming" or selection - some studies/services use receptionists, nurses, ED doctors or GPs; the seniority/experience of these individuals, combined with flexibility of selection criteria (if any) impacts on the performance of the service. Some discussion of context as related to the type and seniority of

staff involved in all steps of the streaming process is warranted. As the authors mention, this distinction is not always apparent in the primary research available, but it is nonetheless important."

Text amended to reflect this (page15).

"Similarly, the gatekeeper role should be interpreted with caution, as professional/institutional governance processes may restrict redirection to external services by nurses, therefore, as with much of the evidence, it may not be comparing like for like."

Added to address this point (page 19).

"With regards satisfaction, GPs are arguably more experienced in managing diagnostic expectation/uncertainty as related to patient satisfaction than (diagnosis driven) ED junior clinicians. This could also contribute to GPs' lower resource utilisation."

We agree but the literature only evidenced improved satisfaction with shorter waiting times. We like the phrase 'Diagnostic driven' ED junior clinician approach (thank you) and have added this (page 16).

"The discussion and conclusions are appropriate, but may now need to take into account some of the points raised above."

Expanded in the discussion section under limitations (page 25).

"Minor: In Strengths and Limitations, 2nd sentence - ..this area, USING methods..."

USING added (page 24).

Reviewer: 2

"This is an interesting and topical paper, particularly given the government policy drive to have a co-located primary care service at all Emergency Departments in England. The authors have followed realist review methodology, using the international literature, to generate theories about how and why general practitioners working in or alongside emergency departments affect: patient attendance and flow; patient experience; patient safety; and the wider healthcare system. Whilst this study design is appropriate to answer the research question, I do have reservations over the broad use of the international literature to answer the research question. Health systems vary considerably across the world, particularly taking into consideration publicly funded versus privately funded health systems, and this will have an impact over the way in which co-located primary care services are implemented / operate. Therefore, there needs to be a greater appreciation of this in the write up of the study. For example, the background to the paper is strongly focused on the UK policy context with no mention of the international perspective."

Please see comments to address reviewer three below to address international relevance and clarify terminology.

"Page 7, line 31: "Six members of the group, including two" could be re-worded. Who are the two?"

Apologies, typo – 'including two' removed (page 8).

"I am uncertain about the relevance of figure 2, perhaps expand on what this means within the main text of the paper. Furthermore, the difference between "primary care type problems" and "low acuity" requires further explanation."

On reflection, we agree with this reviewer that this figure probably does not add to the results or explanation and have therefore removed it.

Reviewer: 3

"Thank you for presenting this scholarly work: 'The effectiveness of primary care service models in or alongside emergency departments: a rapid realist review.' It provides a well written and well developed critique of the roles of general medical practitioners in regard to their working with or within emergency care departments. But the review is restricted to a subgroup of options for models of primary care that have been developed? I would like to make several comments and some suggestions for improvement. Please consider that descriptions need to be explanatory for the international readership – we are not in the UK system and so I suggest your further interpretation needs to be added in some parts."

Many thanks for your review and these comments which we feel have helped presentation for an international audience.

"The title should be revised. The paper does not actually describe primary care service 'models' (GP Co-operatives, walk-in centres, after-hours centres, co-location or otherwise) you restrict your objective to describing GPs working "in or alongside" ED. Unclear- does this include a primary care clinic co-located in the same building as ED? (I think you said 'yes'. If you take the view that you describe general practitioners working "in or alongside", this, then, rather ignores the body of literature about 'models' of care (although I note you list a number of these studies in your tables). Be wary of describing this as 'GP services' or 'primary care services' for this term remains unclear."

The title and emphasis of the manuscript have been amended to reflect the focus specifically on general practitioners working in or alongside emergency departments rather than 'primary care service models'.

"Page 2 Abstract: please be more specific regarding design/content. I have posted some comments on the pdf of the submission. Suggest there is no intervention, please remove."

Thank you – 'intervention' removed and abstract adapted to be more specific about design/content (page 2) .

"Page 3 Abstract line 2-3 Conclusion: "Multiple factors influence the effectiveness of emergency department streaming to primary care services" should be amended to enable understanding that you refer to a general practitioner working within emergency department (not to the broader model of GP services allied to ED)."

Amended to clarify wording as you suggest (page 3).

"Page 3 Abstract Line 9-10: you overstate claims?- " there is little evidence on the safety implications or whether this improves care for the sickest patients." Please amend as 'the sickest patients' are beyond the scope of this review, your GP patient group will be lower acuity, less acutely ill patients?"

Clarified that there is a lack of evidence that streaming non-urgent patients to general practitioners improves care for the sickest patients by freeing up emergency department staff (as suggested by UK NHS policy) (page 3).

"Page 4 line 34: "Three primary care ..." - there is a gap here as you should state that of the patients presenting to ED for emergency care, a number are assessed as less urgent cases and may be triaged to GP for care? Please clarify. Page 4 Line 37-46 please revise to clarify, see notes on PDF"

Clarified to explain how general practitioners are described working in or alongside emergency departments (page 5).

"P6 line2: your citations for Box 1: Glossary of terms(21,23) relate to design, not to definitions. Suggest there needs to be a more detailed definition of Triage and cite the UK policy. 'Streaming' seems to be confounded as I note citations describe a broader view of referral on than mere streaming (you say: "A system to allocate patients to the most appropriate healthcare provider within the emergency department setting"). But the text seems to refer to a broader view of the settings. Lower acuity patients are often referred on to a GP service from ED. This seems missing from your general text and it would be good to state several times that the types of pts that are managed by GPs would be in the lower range of triage 4.5 (pt type not mentioned but this is an important explanatory element)?"

RCEM Definitions of streaming and redirection added to this box to clarify terminology for an international audience (page 7).

"Page 7: your description of the process of synthesis is very helpful."

Thank you for this positive feedback.

"Page 12 line 12: please insert short summary of GP options in this case for streaming – ie, co-located in a clinic, embedded, or sit at front of dept to screen patients? As we are unaware. Please explain " due to the guidance itself"."

Definitions of streaming/redirection clarified in the box (page 7) and differences in guidance clarified (page 15).

"Page 15 line 5 onwards: GP as gatekeeper role. Please add further description here. I am unaware that GPs may be at the forefront of an ED dept- however citation 14 gives some explanation of the context of this 'refer out' situation - that should be added. Is it just 'refer on'? ('UK' is a universal accepted abbreviation)."

Terminology clarified (page 18 and box page 7).

"Page 21 line 18-30 limitations please summarize to reduce repeated text this was all stated in methods. Is not a limitation that you considered only a limited range of GP models- not walk-in centres for example. While I have suggested additions be made to an already lengthy report, there may be some areas where you can reduce the wordiness?"

Paragraph summarised to avoid repetition (page 24).

"Thank you for this comprehensive report. It is difficult to do justice to all your work however I feel that me being an outsider gives me a fresh view of your framework and reporting. Please see both my list of comments and comments on the pdf copy where the connection was more easily made. Please

review punctuation there are too many unnecessary commas that break up the text and perhaps some commas where there should be semicolon? Please review reference list format to be more consistent with the referencing and use of capitalization, etc. I do consider that each reference needs to first give the source or author, however it is up to the publishers. Well done."

Many thanks for the very helpful comments, we hope that the additions to clarify terminology and processes are now more appealing to an international audience. Punctuation and references reviewed.

VERSION 2 – REVIEW

REVIEWER	Shammi Ramlakhan Sheffield Children's Hospital NHS Foundation Trust, UK. University of the West Indies, Trinidad & Tobago.
REVIEW RETURNED	03-Jan-2019

GENERAL COMMENTS	Thank you for your revisions. The manuscript describes useful contextual implications and theories, which policy-makers would do well to read in detail. Unfortunately, in papers such as this, the importance of some of these relatively minor points may be lost. That said, I cannot see how the manuscript can be shortened further, so empathise with the authors... My queries have been addressed in the revision, and along with those from the other reviewers I have no further comments.
--

REVIEWER	Suzanne Ablard Research Associate, University of Sheffield, England
REVIEW RETURNED	02-Jan-2019

GENERAL COMMENTS	The reviewer completed the checklist but made no further comments.
--

REVIEWER	A/Prof Robyn Cant Federation University Australia
REVIEW RETURNED	17-Dec-2018

GENERAL COMMENTS	Thank you for attending to comments of the reviewers in this revision 1. I find the information and enhanced description of context reads well and the added details are very relevant. Thank you, and well done to the first author. You deal with a difficult topic with proficiency. I recommend that the paper is suitable for publication.
---